The usefulness and reliability of English-language YouTube videos as a source of knowledge for patients with familial Mediterranean fever

Coşkun Belkıs Nihan belkisnihanseniz@hotmail.com 1
Yagiz Burcu 1
Giounous Chalil Esra 2
Dalkılıç Ediz 1
Pehlivan Yavuz 1
1 Division of Rheumatology, Department of Internal Medicine, Faculty of Medicine, Uludag University , Bursa , Türkiye
2 Department of Internal Medicine, Faculty of Medicine, Uludag University , Bursa , Turkey
Mitsouras Katherine
Electronic publication date: 2024 Feb 19
Publication date: 2024
Volume: 12
Electronic Location ID: e16857
Received 2023 Aug 24; Accepted 2024 Jan 8
Copyright: ©2024 Coşkun et al.
Copyright year: 2024
Copyright holder: Coşkun et al.
License: This is an open access article distributed under the terms of the Creative Commons Attribution License, which permits unrestricted use, distribution, reproduction and adaptation in any medium and for any purpose provided that it is properly attributed. For attribution, the original author(s), title, publication source (PeerJ) and either DOI or URL of the article must be cited.
License URL: https://creativecommons.org/licenses/by/4.0/

Keywords: Colchicine, YouTube, Usefulness, Treatment of familial Mediterranean fever, Source of knowledge, Reliability, Familial Mediterranean fever

Funding: The authors received no funding for this work.

==============================
Background/Objectives

YouTube is increasingly being used as an educational tool and is a substantial source of information. This study aimed to assess the quality of the most viewed YouTube videos pertaining to familial Mediterranean fever (FMF).

Methods

A search on YouTube was conducted on January 13, 2022, using the keywords: “familial Mediterranean fever treatment,” “familial Mediterranean fever colchicine,” and “familial Mediterranean fever colchicine opacalcium.” Two rheumatologists independently evaluated the relevance and accuracy of the videos. Redundant or irrelevant videos were excluded. The educational value of YouTube videos was assessed using the Global Quality Scale (GQS). Comparative analyses of video parameters across different cohorts were performed. To assess the reliability and quality of the videos, a modified version of the DISCERN scale and the GQS were employed.

Results

Out of the 59 videos reviewed, 43 (72.9%) were of high quality, 10 (16.9%) were of medium quality, and 6 (10.2%) were of low quality. Upon comparing parameters among groups, no significant disparities were observed in terms of daily views, daily favorites, daily dislikes, or daily comments (p > 0.05). GQS scores for usefulness and modified DISCERN scores showed significant differences among groups (p < 0.001). Additionally, both GQS and modified DISCERN scores exhibited moderately negative correlations (r =  − .450 and r =  − .474, respectively) and high statistical significance (p < 0.001 for both) with utility assessment.

Conclusion

YouTube is a valuable repository of high-quality videos for FMF patients. Healthcare providers should guide their patients to high-quality video sources to supplement their educational material.

Introduction

Familial Mediterranean fever (FMF) is an autoinflammatory syndrome distinguished by recurrent fever and serositis. Its clinical manifestations include transient bouts of peritonitis, pleuritis, arthritis, and erythema, typically coupled with fever. Though FMF is predominantly prevalent among Sephardic Jews, Armenians, Turks, and Arabs, it is gradually being identified in diverse populations globally, albeit at a lower frequency (Ozdogan & Uğurlu, 2019; Tufan & Lachmann, 2020). Approximately 90% of FMF patients experience their initial symptomatic episode before the age of 20, with the typical onset age ranging between three and nine years. Instances of FMF development in patients older than 40 years have also been documented, though such occurrences are rare (Ozdogan & Uğurlu, 2019).

Since 1972, colchicine has remained the primary treatment for FMF, as it efficiently mitigates the severity of amyloidosis and prevents seizures in most FMF patients. Proper medication and monitoring can effectively manage most cases of FMF. Colchicine, administered daily, can decrease the intensity and frequency of clinical relapses while delaying the onset of AA amyloidosis, renal failure, and premature death in a majority of patients (Ozen et al., 2016). However, colchicine use in FMF also has potential downsides, including inefficacy due to drug resistance; non-compliance; numerous drug-drug interactions; complications, including potential adverse effects (especially gastrointestinal); and a high risk of toxicity due to its narrow therapeutic range. More FMF patients resistant or intolerant to colchicine are being treated with IL-1 antagonists anakinra and canakinumab, with growing data supporting their efficacy and safety (Tufan & Lachmann, 2020; Ozen et al., 2016; Kharouf, Tsemach-Toren & Ben-Chetrit, 2022).

In our digital era, social media and the internet have become indispensable components of daily life. The internet is an increasingly popular source of health information, with YouTube, a free video platform offering diverse multimedia content, ranking as the world’s second most visited website after Google search (Wikipedia The Free Encyclopedia, 2022; Drozd, Couvillon & Suarez, 2018). YouTube’s accessibility, ease of use, and social networking capabilities make it especially popular for training purposes. However, the quality and reliability of YouTube videos on the topic of FMF is unknown (Kocyigit & Akyol, 2021).

Previous studies have analyzed YouTube video content across various medical categories, but, to our knowledge, no analysis has been conducted specifically on the quality of online video content concerning FMF (Madathil et al., 2015). The objective of this study is to evaluate the accuracy, relevance, and quality of health-related YouTube content on FMF.

Materials & Methods

This study is a descriptive analysis of YouTube videos on the topic of familial Mediterranean fever (FMF). On January 13, 2022, YouTube videos were searched using the keywords: “familial Mediterranean fever treatment,” “familial Mediterranean fever colchicine,” and “familial Mediterranean fever colchicine opacalcium” (Fig. 1). Before conducting the search, the browser’s search history was cleared to minimize the influence of prior internet usage on the search results. Two experienced rheumatologists (BNC and BY) collaboratively selected the search keywords and screened all videos for the selected keywords. The 50 most popular videos for each keyword were included in the study. For each keyword, two experts independently rated the English-language videos. The evaluators were blinded to each other’s ratings during this period. Any discrepancies between the evaluators were addressed by a third observer (EYH). This methodology was informed by several studies that evaluate YouTube video content and credibility by focusing on the top 50 videos (Kocyigit & Akyol, 2021; Zengin & Önder, 2021; Cüzdan & Türk, 2022).

Figure 1 Flowchart.

Data for each video was recorded including video title, web address, origin, duration, upload date, number of views, favorites, rejections, and comments. Daily views, likes, dislikes, and comments were calculated by dividing the total counts by the number of days the video was available on YouTube. Video popularity was determined by calculating the like ratio (likes/[likes plus dislikes] × 100), view ratio (views per day), and the video power index (VPI; like ratio-view ratio/100) (Pamukcu & Izci Duran, 2021). Videos were categorized into two groups based on the upload source: videos uploaded by patients and videos uploaded by healthcare professionals.

Only English-language videos were analyzed. Duplicate videos, irrelevant videos, music videos, and videos lacking audio were all excluded, following the exclusion criteria of previous research. The main objective of this study was to evaluate the reliability, relevance, and quality of the videos using a modified version of the DISCERN instrument and the Global Quality Score (GQS).

Quality assessment

The GQS, frequently used in various YouTube-related studies, was employed to assess the overall quality of the videos (Zengin & Önder, 2021; Elangovan, Kwan & Fong, 2021; Onder & Zengin, 2021). The GQS comprises five questions and a five-point scale (1–5) for evaluating a video’s utility, flow, and quality. Higher scores correspond to higher video fidelity (Table 1).

Table 1 Global Quality Scale (GQS).

1. Poor quality, poor flow, most information missing, not helpful for patients	
2. Generally poor quality and flow, some information given but many important topics missing, of limited use to patients	
3. Moderate quality, suboptimal flow, some important information is adequately discussed but other information is poorly discussed, somewhat useful for patients	
4. Good quality, generally good flow, most relevant information is covered but some topics are not covered, useful for patients	
5. Excellent quality and excellent flow, very useful for patients	

Realibility assessment

Following the original DISCERN written health information assessment tool by (Charnock et al., 1999), a modified DISCERN was employed to assess the reliability of written health information. This assessment tool consists of five questions evaluating clarity, credibility, bias, secondary source citations, and uncertainty management. Each question is answered with a ‘yes’ or ‘no’. Each affirmative response is assigned one point, while negative responses score zero, making the maximum possible score five points, indicating the most reliable health information (Cüzdan & Türk, 2022).

Modified DISCERN dependability instrument (1 point for each affirmative response)

1. Is the video clear, concise, and easy to understand?

2. Are credible sources cited? (Reliable studies, physical therapists and rheumatologists agree)

3. Is the information provided objective and balanced?

4. Are there additional sources of information available to the patient?

5. Does the video address contentious or controversial issues?

Evaluation of usefulness

Two physicians (BNC, BY) assessed each video’s utility, classifying them into mutually exclusive categories based on their assessments (Elangovan, Kwan & Fong, 2021; Tolu et al., 2018). In cases where there was a discrepancy between the two evaluating physicians, a third physician (EYH) served as an arbitrator. The four mutually exclusive categories were:

1. Useful information: Primarily informative videos that provided accurate and informative content about FMF.

2. Misleading information: Videos containing inaccurate or incomplete information about FMF. Videos with both useful and misleading content were classified as misleading.

3. Useful patient opinion: Videos depicting personal experiences or sentiments of an FMF patient.

4. Misleading patient opinion: Videos depicting a patient’s experience that was either ineffective as an educational tool or contained incorrect opinions on the subject.

Statistical analysis

Video data was statistically analyzed using the Statistical Package for the Social Sciences (SPSS) version 22.0 (SPSS Inc., Chicago, IL, USA), and presented as median (minimum to maximum), frequency, and percentage. Data distribution was analyzed using the Shapiro–Wilk test. The Kruskal-Wallis test was used to detect statistically significant differences across multiple categories. Correlations were analyzed using the Spearman test, and the degree of agreement between assessors was determined using the kappa coefficient. The results were interpreted with a significance level of p < 0.05 and a 95% confidence interval.

Ethical approval

As this study evaluated YouTube videos and did not include humans or animals, ethical committee approval was not required.

Results

Out of the 150 videos identified in the initial search, 61 duplicates, 28 off-topic videos, one non-English video, and one video with confidential content were excluded. A total of 59 videos were included in the final analysis, 43 of which were uploaded by healthcare professionals and the remaining 16 by patients sharing their experiences (Fig. 1). Among the 59 videos analyzed, the majority of the main presenters were rheumatologists (n = 23, 38.9%), with 17 being rheumatologists and six being pediatric rheumatologists. Other physicians who contributed to the videos included internists (n = 3), general practitioners (n = 2), and gastroenterologists (n = 2). Additionally, there were two nurses and two pharmacists among the nonphysician healthcare professionals. Five of the videos were hospital information videos or content published by channels such as the American College of Rheumatology YouTube channel and the Food Drug Administration YouTube channel. Four videos were shared by a pharmaceutical company for patient information purposes. The majority of the videos shared by healthcare professionals focused on the pathophysiology, clinical features, treatment, and diagnosis of FMF. Furthermore, eight of the videos included congress presentations by professors who are experts in FMF.

Video duration, number of views, likes, dislikes, and comments for each video are shown in Table 2. The median video length was 4.5 min (range: 0.18–68.56), with a median of 1,170 days (range: 31–24,875) since upload, 1,284 views (range: 25–59,530), and 13 likes (range: 0–359).

Table 2 Basic characteristics of the analyzed videos on FMF (n = 59).

Video data	Median (Minimum–Maximum)	
Number of days on YouTube	1,170 (31–24.875)	
Length of each video (minute)	4.5 (0,18–68,56)	
Number of views	1,284 (25–59.530)	
Number of comments	0 (0–52)	
Number of likes	13 (0–359)	
Number of dislikes	0	
Number of subscribers	3,760 (0–873.000)	
Like ratio	100 (0–100)	
View ratio	1,9 (0.002–64,590)	
Video power index (VPI) score	1,75 (0,002–64,590)	
Modified DISCERN score	4 (0–5)	
GQS score	5 (1–5)	

Interobserver agreement was strong for the GQS (0.926, p = 0.000), the modified DISCERN instrument (0.822, p = 0.000), and video usefulness (1, p = 0.000), as determined by Cohen’s kappa statistic.

Quality assessment

Significant differences were observed in GQS video ratings based on the video upload source, modified DISCERN score, and utility ratings (p < 0.001 for all three; Table 3). High-quality videos had higher modified DISCERN scores, and a greater proportion of videos in the “useful information” group were rated as containing relevant information. The “misleading patient opinion” group had videos of lower quality and reliability.

Table 3 Evaluations of video quality based on the Global Quality Score.

	Low quality
n = 6 (10.2%)	Medium quality
n = 10 (16.9%)	High quality
n = 43 (72.9%)	P K	
Number of days since upload	2,035 (325–2,473)	1,014 (153–3,362)	1,049 (31–24,875)	0.330	
Sources of video upload*	1 (1–1)	2 (1–2)	2 (1–2)	<0.001	
Target audience*	1 (1–1)	1 (1–1)	1 (1–2)	<0.001	
Length of each video (minute)*	1.95 (0.18–16.12)	1.49 (0.47–4.54)	8.16 (1.34–68.56)	<0.001	
Number of views	1,455 (33–4,265)	835 (38–4,963)	1,575 (25–59,530)	0.285	
Number of comments	0 (0–0)	0 (0–0)	0 (0–52)	0.830	
Number of subscribers	6,960 (12–39,800)	5,495 (0–803,000)	3,690 (0–873,000)	0.673	
Number of likes	4.5 (0–30)	9.5 (0–94)	15 (0–359)	0.130	
Number of dislikes	0 (0–0)	0 (0–0)	0 (0–0)	1.000	
Like ratio	100 (0–100)	100 (0–100)	100 (0–100)	0.227	
View ratio	0.74 (0.15–2.26)	1.48 (0.003–64.59)	2.15 (0.002–33.180)	0.120	
VPIβ	0.74 (0–2.26)	1.48 (0–64.59)	1.90 (0.00–33.180)	0.240	
Modified DISCERN score*	0 (0–1)	3 (3–3)	4 (2–5)	<0.001	
Assessment of usefulness*	4 (1–4)	1 (1–3)	1 (1–3)	<0.001	
Notes.

Values of p < 0.05 were accepted as significant and marked bold.

VPI, Video power index

All data are expressed as median (minimum–maximum).

K Kruskal–Wallis test: p < 0.05 significant difference between the groups.

* Mann–Whitney U test →. Sources of video upload: p < 0.001 between high and low groups; p = 0.002 between medium and low groups Target audience: p < 0.001 between high and low groups; p < 0.05 between medium and low groups Length of each video (minute): p < 0.001 between high and medium groups; p < 0.05 between high and low groups Modified DISCERN score: p < 0.001 between high and medium groups; p < 0.001 between high and low groups; p < 0.001 between medium and low groups Assessment of usefulness: p < 0.001 between high and low groups.

Evaluation of video usefulness

Among the analyzed videos, 93.2% (55/59) contained useful information, while 6.8% (4/59) contained misleading information; 44.44% (4/9) of videos produced by independent users contained misleading information, while all videos produced by academic/professional institutions were deemed useful (Table 4). The four misleading videos were uploaded by patients from their own YouTube accounts. No significant difference was observed in the average number of likes/dislikes, comments, or video length between useful and misleading videos. However, useful videos had significantly (p < 0.001) higher reliability and quality scores (DISCERN and GQS) than misleading videos. The characteristics of the YouTube videos, according to their usefulness, are presented in Table 4.

Table 4 Analyses of video characteristics by category of usefulness.

	Useful information
(Group 1)
n = 50 (84.7%)	Useful patient opinion
(Group 3)
n = 5 (8.5%)	Misleading patient opinion
(Group 4)
n = 4 (6.8%)	P K	
Number of days since upload	1,109, 5 (31–24,875)	1,001 (318–3,123)	1,768,5 (325–2,465)	0.783	
Sources of video upload*	2 (1–2)	1 (1–2)	1 (1–1)	0.001	
Target audience*	2 (1–2)	1 (1–1)	1 (1–1)	0.001	
Length of each video (min)	4.88 (0.18–68.56)	3.39 (1–4.54)	6.42 (0.33–16.12)	0.310	
Number of views	1,230 (25–59,530)	1,370 (694–177,87)	1,851 (365–4,265)	0.732	
Number of comments	0 (0–52)	0 (0–0)	0 (0–0)	0.914	
Number of subscribers	4,575 (0–873,000)	191 (104–4,490)	5,500 (12–27,700)	0.232	
Number of likes	13 (0–359)	47 (5–191)	16,5 (2–30)	0.584	
Number of dislikes	0 (0–0)	0 (0–0)	0 (0–0)	1.000	
Like ratio	100 (0–100)	100 (100–100)	100 (100–100)	0.441	
View ratio (views/day)	1.83 (0.002–64.59)	4.24 (0.003–5.690)	1.655 (0.140–2.26)	0.768	
VPIβ	1.73 (0.00–64.59)	4.24 (0.003–5.69)	1.655 (0.140–2.26)	0.808	
GQSβ*	5 (1–5)	5 (3–5)	1 (1–1)	<0.001	
Modified DISCERN score*	4 (1–5)	3 (3–3)	5 (0–1)	<0.001	
Notes.

Values of p < 0.05 were accepted as significant and marked bold.

β VPI: Video power index, GQS Global Quality Score.

All data are expressed as median (minimum–maximum).

K Kruskal–Wallis test: p < 0.05 significant difference between the groups.

* Mann–Whitney U test →; Sources of video upload: p = 0.003 between groups 1 and 3; p < 0.001 between groups 1 and 4 Target audience: p < 0.001 between groups 1 and 3; p = 0.002 between groups 1 and 4 GQS: p < 0.001 between groups 1 and 4; p = 0.016 between groups 3 and 4 Modified DISCERN score: p < 0.001 between groups 1 and 4; p = 0.016 between groups 3 and 4.

Correlation analysis

The correlation analysis revealed a significant positive relationship between the number of views and the number of days since upload, view ratio, like ratio, and VPI (p < 0.001, r = .558; p < 0.001, r = .687; p < 0.005, r = .363; p < 0.001, r = .644, respectively). Positive correlations were also found between video characteristics and ratings for reliability and quality. The correlation between the usefulness rating and the modified DISCERN score and the GQS was moderately negative (r = −.474 and r = −.450, respectively) and highly significant (p < 0.001 for both; Table 5). The correlation between the modified DISCERN score and the GQS was moderately positive (r = .709) and highly significant (p < 0.001).

Table 5 Analysis of the correlations between video quality, reliability, and characteristics.

	Number of views	Number of days since upload	Like ratio	View ratio	VPI β	GQS β score	Modified DISCERN score	Assessment of usefulness	
	p	r α	p	r α	p	r α	p	r α	p	r α	p	r α	p	r α	p	r α	
Number of views	–	1.000	< 0.001	0.558 **	< 0.001	0.36**	<0.001	0.687**	< 0.001	0.644**	0.116	0.207	0.175	0.179	0.622	0.066	
Number of days since upload	–	–	–	1	0.313	0.134	0.712	−0.049	0.821	0.030	0.518	−0.086	0.521	−0.085	0.591	0.071	
Like ratio	–	–	–	–	–	1	0.019*	0.305	< 0.001	0.594**	0.100	0.216	0.206	0.167	0.205	0.167	
View ratio	–	–	–	–	–	–	–	1	< 0.001	0.892**	0.058	0.248	<0.05	0.259*	0.708	−0.050	
VPIβ	–	–	–	–	–	–	–	–	–	1	0.135	0.197	0.119	0.205	0.892	0.018	
GQSβ score	–	–	–	–	–	–	–	–	–	–	–	1	<0.001	0.709**	< 0.001	−0.450**	
Modified DISCERN score	–	–	–	–	–	–	–	–	–	–	–	–	–	1	< 0.001	−0.474**	
Assessment of usefulness	–	–	–	–	–	–	–	–	–	–	–	–	–	–	–	1	
Notes.

Values of p < 0.05 were accepted as significant and marked bold.

β GQS Global Quality Score, VPI: Video power index.

α Spearman p correlation coefficient.

** Correlation is significant at the 0.01 level (2-tailed).

* Correlation is significant at the 0.05 level (2-tailed).

Discussion

To the best of our knowledge, this is the first study that evaluates the content of YouTube videos related to familial Mediterranean fever (FMF). Given the vast amount of unregulated content available on YouTube, we felt a professional review of these videos was necessary, particularly to ascertain their accuracy. Over a period of 12.2 h, we analyzed 59 videos that had accumulated a combined 350,000 views, indicating the popularity of YouTube as a source of information for FMF patients. Our search found that a number of rheumatologists have produced numerous FMF-related YouTube videos for their health information websites.

Patients may prefer a combination of written and verbal information and advice, but healthcare professionals usually only provide written educational materials. YouTube videos have become an established source of information. Appealing, easy-to-understand and peer-reviewed videos can meet the needs of both patients and healthcare professionals (Koo, Kim & Jun, 2021). The internet is a more trusted source of health information than traditional mass media, ranking third after physicians and government health facilities (Ye, 2010). To optimize outcomes in chronic rheumatic diseases such as FMF, patient education is often emphasized as a critical factor (Onder & Zengin, 2021; Ng, Lim & Fong, 2020). The majority of videos (84.5%) in our analysis were uploaded from health information websites. Previous YouTube analyses of different medical conditions found that most medical information videos were uploaded by health professionals (Pamukcu & Izci Duran, 2021; Elangovan, Kwan & Fong, 2021), and patients (Tolu et al., 2018; Rittberg, Dissanayake & Katz, 2016; Delli et al., 2016), which is consistent with the findings of our study.

YouTube is a valuable source of health-related information, but because it is a free platform, the diversity of contributors and the absence of a filtering mechanism can lead to the dissemination of misinformation. A significant number of studies have evaluated YouTube video content across various disciplines. While some studies considered YouTube a reliable source of information (Pamukcu & Izci Duran, 2021; Onder & Zengin, 2021), others found that it could provide misleading information to patients (Mangan et al., 2020; Baydilli & Selvi, 2022; Selvi, Baydilli & Akınsal, 2020). In our study, 72.9% of the videos were of high quality, aligning with the findings of other studies where the majority of videos were of average to high quality (Pamukcu & Izci Duran, 2021; Onder & Zengin, 2021). We attribute the high percentage of high-quality videos in our study to the fact that our video sources were primarily health information websites, where the majority of content was presented by physicians.

The majority of videos in this study were rated as useful, of average quality, and reliable using the modified DISCERN instrument and the Global Quality Score (GQS) rating system. This is consistent with other studies assessing the quality and credibility of YouTube health information across various rheumatologic conditions (Pamukcu & Izci Duran, 2021; Onder & Zengin, 2021; Zengin & Onder, 2020). Notably, while useful videos had higher GQS and DISCERN ratings, there was no significant difference in viewer interaction characteristics (such as views, likes, and comments per day) between useful and misleading videos (Pamukcu & Izci Duran, 2021; Delli et al., 2016). However, previous research has shown that interaction characteristics of informative and misleading videos are different. For example, Garg et al. (2015) showed that misleading videos attract more viewers. Similarly, Mangan et al. (2020) evaluated YouTube videos on strabismus and found that users mostly preferred low-quality videos. Consequently, the number of daily views, likes, dislikes, and comments a YouTube video has does not accurately reflect the accuracy of its content, indicating that viewers have difficulty distinguishing between relevant and irrelevant information. To prevent the spread of false information, doctors and healthcare providers should upload more medical content with reliable and accurate data. It is important to inform individuals that videos featuring medical advertisements should not be blindly accepted without first consulting nonprofit physicians. Otherwise, videos with misinformation may gain popularity (Baydilli & Selvi, 2022).

FMF is a distressing condition characterized by episodes of fever and significant symptoms. It is typically inherited in an autosomal recessive manner, with the majority of patients experiencing their first attack before age 20. FMF is therefore considered a childhood disease, and colchicine prophylaxis is the mainstay of treatment (Livneh & Langevitz, 2000). Although colchicine treatment has significantly improved the prognosis of FMF patients by reducing episodes of fever and preventing amyloidosis, caregivers continue to bear a substantial burden (Koşan et al., 2019). Patients and their families may require additional information about the disease, considering its onset in childhood, genetic predisposition, and lifelong treatment requirement. YouTube, with its 24/7 availability of information, could supplement advice or information given during limited visits to the doctor.

Limitations

This study is cross-sectional, and the dynamic nature of YouTube, with its content continually updating, imposes certain limitations. To mitigate the effects of this restriction, we utilized snapshot analysis, as in previous research, to record the videos on a playlist (Selvi, Baydilli & Akınsal, 2020). The analysis of the 50 most popular videos (a total of 150 videos) pertaining to each research topic may also be regarded as a constraint in our study. This number was determined by previous research indicating that the majority of YouTube users only peruse the initial few pages of search results (Mangan et al., 2020). Although our analysis is comprehensive and includes a large number of videos, individuals may view only a few videos, which may not always be applicable to real-life scenarios. Also, our analysis was confined to English-language videos, which might limit the scope of our findings. However, English remains the preferred language for online information access. Additionally, despite their frequent usage, tools like DISCERN and GQS could be affected by reviewer bias, emphasizing the need for a more objective evaluation index to enhance the accuracy and dependability of medical data.

Conclusions

Our findings suggest that patients looking for reliable information on FMF on YouTube stand a good chance of finding it, especially if they focus on videos created by health professionals, particularly those linked with academic institutions.

Supplemental Information

Supplemental Information 1 FMF YouTube Data

Additional Information and Declarations

Competing Interests

Author Contributions

Data Availability

The authors declare there are no competing interests.

Belkıs Nihan Coşkun conceived and designed the experiments, performed the experiments, analyzed the data, prepared figures and/or tables, authored or reviewed drafts of the article, and approved the final draft.

Burcu Yagiz conceived and designed the experiments, performed the experiments, analyzed the data, prepared figures and/or tables, authored or reviewed drafts of the article, and approved the final draft.

Esra Giounous Chalil analyzed the data, authored or reviewed drafts of the article, and approved the final draft.

Ediz Dalkılıç analyzed the data, authored or reviewed drafts of the article, and approved the final draft.

Yavuz Pehlivan analyzed the data, authored or reviewed drafts of the article, and approved the final draft.

The following information was supplied regarding data availability:

The raw data is available in the Supplementary File.

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
