# Peer review of "The usefulness and reliability of English-language YouTube videos as a source of knowledge for patients with familial Mediterranean fever"

_PeerJ, doi:10.7717/peerj.16857_

## Round 0.1 · original submission · Major Revisions

Your manuscript was considered interesting by the reviewers however they had a number of concerns that need to be addressed. They would like you to state that even though your analysis is comprehensive and included a large number of videos, it is not necessarily applicable to real-life scenarios where individuals might only view a few videos. Additionally, they suggest that you address in the discussion section the discrepancy between your conclusion that YouTube is a valuable repository of high-quality information for FMF and current literature that concludes that it is not a reliable source of medical information. The reviewers would also like you to consult with a statistician regarding whether solely using the Shapiro-Wilke test for normality is sufficient. Lastly ,they request that you provide more information on the content (subtopic) of the video as well as the specific specialty/professional role for creators that are healthcare professionals.



Please, submit a detailed rebuttal which shows where and how you have taken all comments and suggestions into consideration. If you do not agree with some of the reviewers’ comments or suggestions, please explain why. Your rebuttal will be critical in making a final decision on your manuscript. Please, note also that your revised version may enter a new round of review by the same or by different reviewers. Therefore, I cannot guarantee that your manuscript will eventually be accepted.

**Language Note:** The review process has identified that the English language must be improved. PeerJ can provide language editing services - please contact us at [email protected] for pricing (be sure to provide your manuscript number and title). Alternatively, you should make your own arrangements to improve the language quality and provide details in your response letter. – PeerJ Staff

Reviewer 1 ·

Basic reporting

The authors evaluate the usefulness and reliability of English-language YouTube videos as a source of knowledge for patients with familial Mediterranean fever.
Please improve the grammatical and spelling error.
Please add the reference
Mangan MS, Cakir A, Yurttaser Ocak S, Tekcan H, Balci S, Ozcelik Kose A. Analysis of the quality, reliability, and popularity of information on strabismus on YouTube. Strabismus. 2020 Dec;28(4):175-180. doi: 10.1080/09273972.2020.1836002. Epub 2020 Oct 19.

They concluded that Youtube videos on FMF while having reliable publishers are high in quality.
Well written paper and it is original enough to be published in this journal after the minor revision.

Experimental design

.

Validity of the findings

.

Reviewer 2 ·

Basic reporting

It would be more appropriate to define the contents of the videos whether they are for pathophysiology, for medical therapy, physical therapy etc.
And give detail about health care profesşionals

Experimental design

no comment

Validity of the findings

no comment

Reviewer 3 ·

Basic reporting

In the last few years, there have been many publications investigating the usability of Youtube videos as a reliable source of information on different medical topics. Most publications have concluded that most viewed YouTube videos are not a reliable source on medical topics. However, this paper emphasizes that YouTube is a valuable repository of high-quality videos for FMF patients. As mentioned in the discussion, because it is a free platform, the diversity of contributors and the absence of a filtering mechanism can lead to the dissemination of misinformation. Therefore, I believe that it may be necessary to mention some current literature data in the discussion section regarding these contradictory findings (e.g.; doi: 10.1016/j.urology.2020.06.082 doi: 10.1038/s41443-021-00454-3)

Experimental design

The authors reviewed the most popular 150 videos. Unfortunately, the average patient/lay person, will not review that many and rely on the first 2-3 and trust that, if not sponsored sites. That number of views or position of the video in rank order suggests accuracy or more likely popularity. The authors need to state that even though they looked at a robust number, real life may be less investigative. Ideally the most accurate and best understood and validated video should be the site that pops up/listed to view.

Validity of the findings

It is not sufficient to analyze the normality test of continuous variables only with the Shapiro-Wilk test. I suggest reviewing this assessment by consulting a statistician.

---

## Round 0.2 · accepted · Accept

Thank you for thoroughly addressing all the reviewers' comments and thus greatly improving your manuscript.

Reviewer 1 ·

Basic reporting

nice revised paper. thank you

Experimental design

nice revised paper. thank you

Validity of the findings

nice revised paper. thank you

Additional comments

nice revised paper. thank you

Reviewer 3 ·

Basic reporting

In my opinion, the paper has become valuable and it can contribute to the literature after the revision made according to the reviewers' recommendations, so I congratulate the authors for their efforts.

Experimental design

No comment

Validity of the findings

In my opinion, the paper has become valuable and it can contribute to the literature after the revision made according to the reviewers' recommendations, so I congratulate the authors for their efforts